# Comparison of Crown Volume Increment in Street Trees among Six Cities in Western Countries and China

Chenbing Guo [1], Yonghong Hu [2,3], Jun Qin [2], Duorun Wu [1], Lin Xu [1] and Hongbing Wang [1,*]

1   Shanghai Engineering Research Center of Plant Germplasm Resources, College of Life Sciences, Shanghai Normal University, No. 100 Guilin Road, Xuhui District, Shanghai 200234, China; gcb7917@163.com (C.G.); wdr000103@163.com (D.W.); xl0087@126.com (L.X.)
2   Shanghai Chenshan Botanical Garden, No. 3888 Chenhua Road, Songjiang District, Shanghai 201602, China; huyonghong@csnbgsh.cn (Y.H.); qinjun03@126.com (J.Q.)
3   Shanghai Chenshan Plant Science Research Center, Chinese Academy of Sciences, No. 3888 Chenhua Road, Songjiang District, Shanghai 201602, China
*   Correspondence: whb0236@shnu.edu.cn; Tel.: +86-21-57122689

**Abstract:** The tree crown volume (CV), as a major indicator in the evaluation of ecological environment quality, can assess the health and carbon sequestration of urban trees. In this study, a new low-cost method, the plane calculation of angle disparity (PCAD), was employed to obtain the CV in China using satellite images from Google Earth. Meanwhile, primary data on street trees from four Western cities were acquired from online datasets. Nonparametric statistical methods showed no significant difference in CV per street tree between Beijing and Shanghai in China, ranging from 10 to 150 $m^3$, almost one-seventh of that in the four cities (Paris and London in Europe and Los Angeles and Seattle in America). The CV of *Platanus acerifolia* in Paris and London exhibited values five times higher than those in Beijing and Shanghai. The annual crown volume increment (CVI) was less than 5 $m^3$ in Beijing and Shanghai, significantly lower than in Seattle (66.55 $m^3$). The purpose of the research was to verify the operability of the PCAD and compare the CVI in different cities all over the world, providing new ideas for urban tree management and carbon sequestration evaluation and a basis for government decision making in areas with a low CVI.

**Keywords:** urban trees; crown volume (CV); crown volume increment (CVI); plane calculation of angle disparity (PCAD); virtual survey; carbon sequestration



## 1. Introduction

Increasingly, cities worldwide are becoming hotter because of climate change and urbanization [1]. Heat has become a concern for cities everywhere. Fortunately, urban green infrastructure can mitigate these impacts and provide some relief from the urban heat-island effect [2,3]. Street trees are one of the closest and most well-known green infrastructures for urban dwellers, and their importance in the ecological environment has been recognized [4,5]. Providing substantial green cover in built-up areas with little green space, street trees generate various ecosystem services, such as improving microclimates by shading buildings, slowing wind speeds, cooling air, and providing diverse habitats for organisms [6–9]. Trees store carbon as they grow and reduce energy consumption in buildings through transpiration and shading, reducing carbon emissions [10,11]. However, their carbon sequestration has received less attention in research [12]. To achieve these values, it is essential to ensure that trees are maintained in a healthier state. Thus, a comprehensive and accurate quantification of street trees provides valuable information for urban ecological construction, which is a necessary condition for the assessment of carbon sequestration in urban trees [13]. The shape and size of tree crowns, influenced by a tree's growing condition, play a pivotal role in quantifying individual street trees [14]. A healthy tree with a larger crown diameter usually has a larger quantity of leaves, which is positively

related to carbon sequestration [15]. Crown assessment can be used as an indicator of health conditions in forest trees [16]. Air pollution capture, rainfall retention, and shade provision are directly related to the surface area of the leaves and branches within a tree crown [17]. Thus, there is a strong dependence on healthy foliage and intact tree crowns in order to sustain and maximize the ecosystem services provided [17]. The crown volume (CV) is defined as the three-dimensional volume of space occupied by plant leaves and stems. As a three-dimensional greening index, it guides the human perception of greenery from two-dimensional areas to three-dimensional spaces. It can more accurately reflect the rationality of the spatial composition of plants and the ecological benefits provided by a greenspace system [18]. CV becomes the key to the estimation of carbon stock [19]. However, cities can compromise this because urban trees are often susceptible to pruning or other artificial disturbances, affecting the crown volume [17].

Research on the concept of CV can be traced back to the 1980s, which mainly focused on the measurement methods of CV and the ecological benefits of urban green space [20,21]. On a global scale, CV research involves studying the services provided by urban trees in a certain area or using models to assess urban forest structure and ecosystem services in multiple cities [15,22]. Allometric equations are used as a nondestructive method for estimating urban tree biomass and carbon storage [23]. LiDAR data are a valid source for estimating the CV of individual street trees. For example, waveform LiDAR provides information spanning from the top of a canopy down to the ground, vividly showing the complexity of urban vegetation [24]. Terrestrial laser scanning (TLS) has been used to produce high-resolution 3D vegetation maps. Field measurements at larger sites have also demonstrated that TLS data are more consistent and accurate than manual methods [25,26]. Airborne laser scanning (ALS) can measure canopy height and estimate canopy density, while ground-based calibration is necessary to reduce bias [27]. In recent years, it has been more convenient to acquire high-resolution aerial image data to dramatically improve the efficiency of the ecological evaluation of street trees [28,29]. The accuracy of the point cloud model of a single tree established using Unmanned Aerial Vehicle (UAV) tilt aerial photography technology can meet the requirements of calculating tree height and CV in forest surveys [30,31]. It has lower data collection costs, higher spatial resolution and improves the accuracy of CV estimation [32–36].

However, CV calculation methods have problems, such as long calculation times and high costs [37]. More research interest should be paid on the improvement of the calculation method of CV and its application. Based on the plane calculation of angle disparity (PCAD), a new calculation method for CV, this study tried to calculate the crown volume increment (CVI) of different street tree species in cities [38]. The PCAD is a method that uses the inclination angles of satellite images to calculate tree height according to the reference object and the CV using the indicators of tree height, stem height, and crown diameter. The CVI, the difference in the CV of one individual tree during a certain period, provides information for future streetscape management with an improved knowledge of species-specific growth rates and vitality [39]. Previous studies have used the allometric equation to compare the crown size of urban saplings in the northeastern USA after five and fifteen years of planting and found the growth rates of the ten most planted species and a strong relationship between CV and stem size [40]. The CVI can be used to explore the effect of the spatial distribution of trees and the differences between different social statuses or productivity [41,42]. It is possible to quantify structural changes and the competitive status of trees in space and time. In addition, the size of the crown and the CVI are also inseparable from exploring the ecological benefits of urban trees [43]. For example, trees with dense (high LAI) and wide canopies are beneficial for mitigating urban heat [44]. The CVI can also reflect the damage to *Picea abies* caused by environmental pollution [45]. Nevertheless, it remains challenging to obtain the CVI, and few studies have focused on the CVI in different cities [46]. Despite this, there are few studies on CV dynamic changes, especially using a method that is low-cost and has a fast operation speed. This highlights the necessity and innovativeness of this study.

This study uses the following two data sources to calculate the CV and CVI. The first is to use the plane calculation of angle disparity (PCAD) to obtain data with the help of software such as Google Earth Pro (Google Inc., Mountain View, CA, USA) and ArcGIS (Environmental Systems Research Institute, Inc., Redlands, CA, USA). In the PCAD, it can be found that the three-dimensional index tree height can be calculated based on two two-dimensional satellite images, which was applied to the CV calculation. It has the advantages of low costs, fast operation, and a large applicable area. The second is open and free online datasets. Based on the above data sources of the PCAD and open datasets, the following questions will be explored: first, how the CVIs of urban street trees change dynamically; second, what are the differences in the CVs of different cities worldwide? The results offer a way of assessing the CVI related to carbon sequestration in urban trees and promote knowledge exchange among cities with different management styles.

## 2. Materials and Methods

### 2.1. Study Area

This study was conducted in the following six cities: Beijing, China (16,410 km$^2$; 39°56′ N, 116°20′ E; population 21,893,000); Shanghai, China (6340 km$^2$; 31°14′ N, 121°29′ E; population 24,871,000); Paris, France (105.4 km$^2$; 48°52′ N, 2°25′ E; population 2,240,000); London, England (1577 km$^2$; 51°30′ N, 0.1°5′ E; population 9,541,000); Los Angeles, the USA (1215 km$^2$; 34°03′ N, 118°15′ W; population 4,087,000); Seattle, the USA (369.2 km$^2$; 47°38′ N, 122°2′ W; population 3,433,000). The two Chinese cities were selected because of the largely progressed planning and construction of urban green spaces. The four cities in Western countries were selected based on the available online street tree data. All the sampled cities are distributed widely in Asia, Europe, and North America in the northern hemisphere. Moreover, the urban planning of these six cities is mature and has a particular scale of urban street tree planting, which has typicality and representativeness.

### 2.2. Virtual Research Data Acquisition

In many cities, specialized agencies have been conducting a census of urban street trees since as early as ten years ago, including the geographical location, ID number, scientific name, DBH (cm), height (m), age, and planting position of each street tree. The data are publicly available on local government or specialized agencies' websites. Four of the six cities, Paris, London, Los Angeles, and Seattle, used data on the internet from open datasets. If the content of the open dataset is not comprehensive enough, it can be obtained from Google Street View (https://www.google.com/maps/, accessed on 12 June 2023.) [47,48]. When acquiring data using Street View, the number of sampling trees was below 100 when the total number of trees was lower than 100, and all trees were measured; the sampling trees should be randomly selected for measurement, and the sampling number was 100 when the total number of trees was more than 100.

The street tree surveys in Beijing and Shanghai were conducted through the PCAD where there was no publicly available street tree data online, and sample surveys were conducted based on the sample plots, which were related to the city's size and the road network layout. The sample plots were set up as follows (Figure 1): 1°, radial lines in four cross-section directions were laid out from the urban center; 2°, rings were drawn at 1 km intervals, defining the urban center as the center of the circles; and 3°, circle-shaped sample plots were set up with 1 km diameters at 3 km intervals, where the circle centers were intersected with the radial lines, and each plot was located between two adjacent rings [38]. Each plot is a circle with a diameter of 1 km and an area of approximately 3.14 km$^2$. There are 13 sample plots and 714 sample trees in Shanghai and 15 sample plots and 1738 sample trees in Beijing. After determining the sample plots and sample trees, Google Earth Pro was used to download the satellite images, and ArcGIS was used for image interpretations. We downloaded satellite images of Beijing and Shanghai (Table 1). The images were combined using Photoshop (Adobe Systems Incorporated, San Jose, CA, USA). After that, the coordinates of the same key points in the same plots were obtained

on the two images, and the tree height was calculated based on the coordinates. Crown diameter can be measured directly in ArcGIS. Additionally, the tree species and stem height were determined with the help of Street View [47–49]. Stem height, defined as the height from the base of the tree trunk to the first first-level branch, was subtracted from the tree height to obtain the crown height.

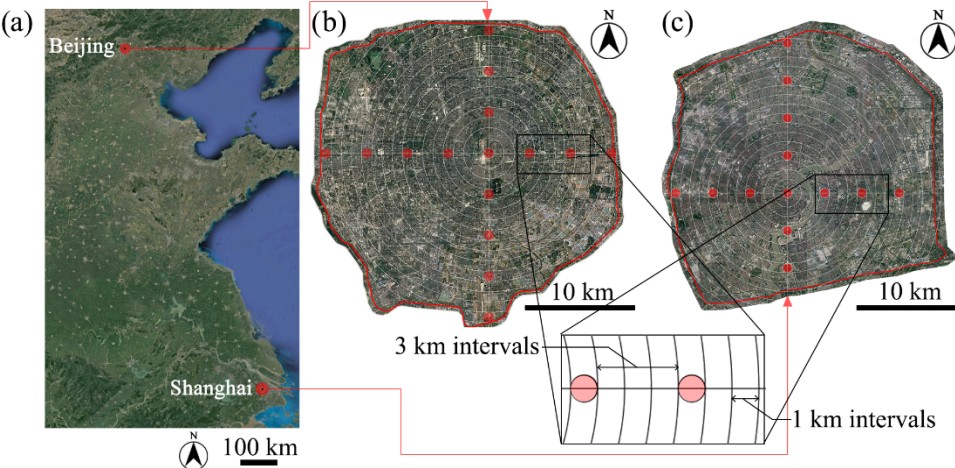

**Figure 1.** Schematic diagram of the two sample cities in China and the layout of the sample sites. (**a**) Sample sites in part of a China map (image downloaded from Google Earth Pro). (**b**) Beijing, China. (**c**) Shanghai, China. A red dot is a sample site; a red circle or line is the fifth ring road or the outer ring road, respectively.

**Table 1.** The dates of data sources in the six cities.

| City | Data Source | Date of Data |
|---|---|---|
| Beijing, China | satellite images | 4 May 2010<br>27 September 2010<br>27 October 2010<br>8 November 2010<br>3 August 2020<br>28 August 2020<br>14 April 2021<br>14 November 2021 |
| Shanghai, China | satellite images | 2 April 2019<br>13 February 2010<br>22 July 2010<br>13 August 2010<br>12 November 2010<br>9 November 2019<br>10 December 2019<br>20 August 2020<br>29 May 2021<br>26 September 2021<br>14 November 2021 |
| London, UK | the London government website's street tree map (https://apps.london.gov.uk/street-trees/, accessed on 11 August 2022.) | 2021 |
| Paris, France | the Paris Open Data website (https://opendata.paris.fr/, accessed on 11 August 2022.) | 2021 |
| Seattle, USA | the Seattle Department of Transportation (www.seattle.gov, accessed on 20 June 2022.) | 2015 |
| | the Seattle Open Data website (https://public.tableau.com/app/profile/city.of.seattle.transportation/viz/SDOTTreeSelector/Dashboard, accessed on 15 July 2022.) | 2022 |
| Los Angeles, USA | the statistics of TreePeople (https://www.treepeople.org/, accessed on 8 August 2022.) | 2021 |

### *2.3. Calculation and Analysis of CV of Street Trees*

The calculation of CV for each tree was based on its crown diameter, crown height, and the "CV equation". The CV calculation process is shown in Figure 2. The crown diameter and height of each tree were obtained from virtual research. In Table 2, The "CV equation" was based on previous studies, and each tree species was matched with a regular geometry [50–53]. With the indicators of crown diameter and crown height, we can calculate the CV (V) per street tree based on crown shape. The annual tree height increment (aH), annual crown diameter increment (aW), and annual CVI (aV) were also calculated, and they were obtained by subtracting the data with more than five-year intervals and dividing by the year. In this study, the calculation of aH, aW, and aV was based on the data spanning the ten years between 2010 and 2020 in Beijing and Shanghai (e.g., $aV = (V_{2020} - V_{2010})/10$), and those spanning the seven years between 2015 and 2022 in Seattle (e.g., $aV = (V_{2022} - V_{2015})/7$). For plots from Beijing and Shanghai without satellite images of 2010 and 2020, 2009 data instead of 2010 data and 2019 and 2021 data instead of 2020 data were used. The virtual survey of street trees in the sample cities at different times revealed changes in street tree species. The changes here referred to tree replacement or transplanting, which could be observed in satellite images or datasets. The aH, aW, and aV of the remaining street trees were analyzed by excluding the changed trees.

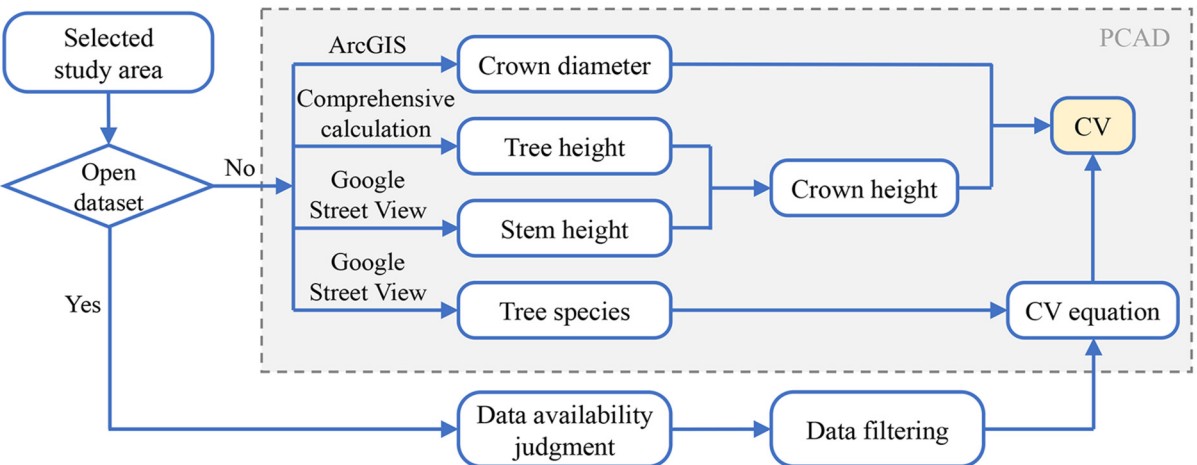

**Figure 2.** A flow chart of CV calculation.

**Table 2.** Formulas used for CV calculation.

| Geometry of Crown Shape | CV Equation |
|---|---|
| Sphere | $\pi x^2 (y - z)/6$ |
| Columnar | $\pi x^2 (y - z)/4$ |
| Cone | $\pi x^2 (y - z)/12$ |
| Global fan style | $\pi\left(2y^3 - y^2 \times \sqrt{4y^2 - x^2}\right)/3$ |

$x$ is the crown diameter; $y$ is the tree height; and $z$ is the stem height.

### *2.4. Data Analysis*

All statistical analyses were performed using SPSS 26.0 statistical software (IBM Corp., Armonk, NY, USA). Before choosing statistical criteria, all data were checked for normality distribution, and homogeneity of Variances. For these data (aH), a one-way ANOVA test was used, and the least significant difference (LSD) method was used for post hoc multiple comparisons. Statistical significance was defined as $p < 0.05$. For the non-normal data (H, W, CV, aW, and aV), nonparametric statistical methods were applied, including the Kruskal–Wallis test. We analyzed the CV, tree height, and crown diameter and calculated their mean variables to compare the differences between different tree species and cities. Values in parentheses represent 95% confidence intervals.

## 3. Results

### 3.1. Basic Characteristics of Street Trees in Cities

By comparing the characteristics of 35 street tree species, including 5–7 species commonly used in each city (Table 3), we found some similarities and differences. The highest-frequency genus was in the *Aceraceae* (20.00%), followed by the *Platanaceae* (11.42%). The evergreen species occurred with a low frequency of 11.42% only in Shanghai and Los Angeles, where latitudes are relatively low. Among all twenty-seven tree species, the top three species with the highest CVs were *Acer saccharum*, *Cinnamomum camphora*, and *Fraxinus americana*, while the top three tree species with the lowest CVs were *Populus tomentosa*, *Salix babylonica*, and *Metasequoia glyptostroboides*. The characteristics of the same tree species were different between the two Chinese cities and the four Western cities, i.e., the latter had a larger height and crown diameter than the former. For example, the 95% confidence interval for the tree height of *P. acerifolia* in Paris was 12.78~12.90 m, and in Shanghai, it was 5.85~6.61 m. The average height of street trees was 8.05 m in the two Chinese cities, 10.33 m in Paris and London, and 14.96 m in Los Angeles and Seattle. Among them, the average tree height (Figure 3a) in Seattle was 16.76 m, which was significant ($\chi^2 = 21.217$, $p < 0.05$). In terms of the average crown diameter (Figure 3b), the street trees in Beijing (3.23~5.14 m) and Shanghai (3.34~6.74 m) were smaller than those in Los Angeles (7.89~15.46 m) and Seattle (5.52~13.88 m). Similarly, there were significantly smaller CVs and lower carbon sequestration per street tree in China.

**Table 3.** Basic information on the CVs of street tree species in the six sampled cities during 2021–2022.

| City | Species | Number | H (m) | Solid Geometry Shape of Tree Crown | W (m) | CH (m) | CV (m³) |
|---|---|---|---|---|---|---|---|
| Beijing, China | *Sophora japonica* | 1037 | 7.07 | sphere | 5.46 | 4.78 | 74.62 |
| | *Populus tomentosa* | 355 | 6.84 | sphere | 3.94 | 4.20 | 34.02 |
| | *Fraxinus chinensis* | 192 | 9.60 | sphere | 3.49 | 7.18 | 45.85 |
| | *Salix babylonica* | 78 | 5.07 | sphere | 4.26 | 3.17 | 30.06 |
| | *Platanus acerifolia* | 76 | 8.01 | columnar | 3.77 | 5.78 | 64.41 |
| Shanghai, China | *Cinnamomum camphora* | 504 | 9.11 | sphere | 5.92 | 6.02 | 110.38 |
| | *P. acerifolia* | 179 | 6.23 | columnar | 5.61 | 3.41 | 84.08 |
| | *Metasequoia glyptostroboides* | 12 | 8.11 | cone | 2.84 | 5.54 | 11.71 |
| | *Sapindus saponaria* | 8 | 8.09 | sphere | 6.65 | 5.70 | 132.25 |
| | *Ginkgo biloba* | 6 | 9.53 | cone | 3.16 | 7.02 | 18.39 |
| | *Acer buergerianum* | 5 | 10.94 | sphere | 6.07 | 7.37 | 142.02 |
| London, UK | *P. acerifolia* * | 856 | 14.34 | columnar | 8.17 | 11.29 | 591.99 |
| | *F. excelsior* | 704 | 8.73 | sphere | 4.83 | 5.75 | 70.20 |
| | *A. campestre* | 546 | 6.47 | sphere | 4.61 | 3.32 | 36.94 |
| | *A. platanoides* | 416 | 9.22 | sphere | 5.99 | 6.07 | 113.83 |
| | *Prunus avium* | 195 | 6.65 | sphere | 4.45 | 4.27 | 44.26 |
| | *Tilia platyphyllos* | 21 | 11.55 | sphere | 5.83 | 8.97 | 159.80 |
| Paris, France | *P. acerifolia* * | 35,055 | 12.84 | columnar | 9.30 | 9.46 | 642.50 |
| | *Aesculus hippocastanum* | 18,476 | 12.60 | sphere | 6.10 | 9.73 | 189.45 |
| | *S. japonica* | 10,609 | 9.99 | sphere | 6.28 | 6.45 | 133.30 |
| | *T. tomentosa* | 7323 | 10.58 | sphere | 8.12 | 7.27 | 251.15 |
| | *A. platanoides* | 5204 | 9.55 | sphere | 7.07 | 7.01 | 183.12 |
| | *A. pseudoplatanus* | 4708 | 12.03 | sphere | 8.30 | 8.95 | 323.20 |
| | *T. europaea* | 533 | 9.78 | sphere | 8.64 | 6.29 | 245.74 |
| Seattle, USA | *A. nigrum* 'Green Column' | – | 18.29 | columnar | 8.08 | 14.78 | 757.77 |
| | *F. americana* 'Empire' | – | 16.76 | columnar | 9.28 | 14.21 | 960.12 |
| | *G. biloba* 'Princeton Sentry' | – | 15.24 | columnar | 5.78 | 13.13 | 344.31 |
| | *A. saccharum* 'Bonfire' | – | 18.29 | sphere | 14.39 | 15.00 | 1625.31 |
| | *Ulmus parvifolia* 'Emer II' | – | 15.24 | sphere | 11.45 | 12.15 | 834.11 |
| Los Angeles, USA | *Pinus pinea* | – | 18.29 | global fan style | 12.19 | 14.31 | 584.66 |
| | *Podocarpus macrophyllus* | – | 10.67 | sphere | 8.33 | 7.80 | 283.18 |
| | *G. biloba* | – | 17.53 | cone | 12.19 | 13.66 | 531.43 |
| | *Jacaranda mimosifolia* | – | 9.91 | sphere | 9.91 | 6.87 | 352.78 |
| | *Cercis canadensis* | – | 9.14 | sphere | 9.14 | 6.79 | 297.44 |
| | *C. camphora* | – | 15.24 | sphere | 18.29 | 11.60 | 2031.37 |

* Also known as *Platanus × hispanica*.

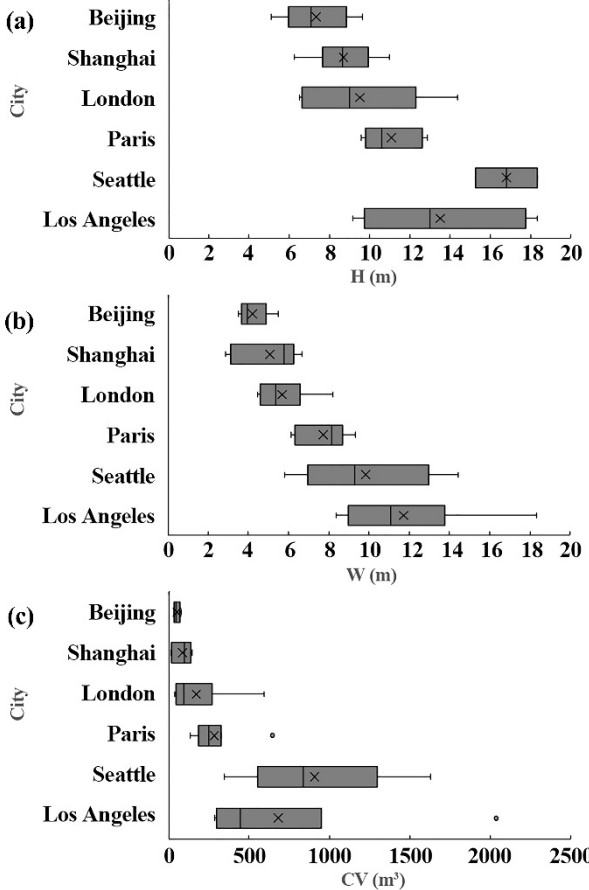

**Figure 3.** Average tree height, crown diameter, and CV in six cities. (**a**) Comparisons of average tree height in six cities. (**b**) Comparisons of average crown diameter in six cities. (**c**) Comparisons of average CV in six cities.

For the crown type, the most abundant was the sphere, followed by columnar, cone, and global fan style. The average tree height for spherical crowns was 10.10 m; columnar was 13.10 m; and cone was 11.72 m. The average crown diameter of the spherical crowns was 7.40 m; columnar was 7.14 m; and cone was 6.07 m. The average CV of the spherical crowns was 322.68 $m^3$; columnar was 492.17 $m^3$; and cone was 187.18 $m^3$. The crowns with sphere and columnar shapes had larger crown sizes than the cone-shaped crowns. However, there was no significant difference in the CV for the trees with different crown shapes, while the trees with the same crown shape differed in CV by species, reflecting the combined effects of the tree species and crown shape on the CV. And the crown shape of global fan style was not included in the discussion because there was only one sample species.

H—tree height; W—crown diameter; CH—crown height; V—crown volume (CV). The data for Paris in 2021 were sourced from the Paris Open Data website (https://opendata.paris.fr/, accessed on 11 August 2022.). The data for London in 2021 were sourced from the London government website's street tree map (https://apps.london.gov.uk/street-trees/, accessed on 11 August 2022.). The data for Los Angeles in 2021 were sourced from the statistics of TreePeople (https://www.treepeople.org/, accessed on 8 August 2022.), but it was processed data, making it impossible to determine the number of street trees. The data for Seattle in 2022 was sourced from the Seattle Open Data website (https://public.tableau.com/app/profile/city.of.seattle.transportation/viz/SDOTTreeSelector/Dashboard, accessed on 15 July 2022.), but it was processed data, making it impossible to determine the number of street trees.

### 3.2. Difference in CV in Cities

The average CV value per street tree standing in six cities ranged from a minimum of 11.71 m$^3$ to a maximum of 2031.37 m$^3$. There was a relatively obvious positive relationship between the CV and crown diameter. The CV increased with an increase in the crown diameter at a similar tree height. For example, the comparison between *Acer platanoides* and *Tilia europaea* showed that their crown diameters were 7.07 m and 8.64 m, respectively, and their CVs were 183.12 m$^3$ and 245.74 m$^3$, respectively. The average CV per street tree in the two Chinese cities was 67.98 m$^3$, which was smaller than that of 482.83 m$^3$ in the four Western cities. In particular, the CVs in Shanghai (23.93~142.35 m$^3$) and Beijing (25.88~73.71 m$^3$) were significantly lower than the CVs in Seattle and Los Angeles ($\chi^2$ = 25.141, $p < 0.05$) (Figure 3c).

From the perspective of different tree crown shapes, the average CVs of the sphere and columnar crowns in the two Chinese cities were 81.32 m$^3$ and 74.24 m$^3$, respectively. On the contrary, the average CVs of the sphere crowns and columnar crowns in the four Western cities were 422.07 m$^3$ and 659.34 m$^3$, respectively. It can be found that the CV of the same tree crown shape in Europe and America was 5–8 times that of China. In addition, there were also differences in the CVs of the same tree species among the different cities. For instance, the tree height, crown diameter, and CV of *P. acerifolia* in Beijing and Shanghai were slightly smaller than those in London and Paris.

### 3.3. Difference in CVI in Cities

The average annual increment in tree height, crown diameter, and CV in Beijing, Shanghai, and Seattle can be seen in Figure 4. The CV and crown diameter had a strong correlation. The average annual crown diameter increment (aW) and average annual CVI (aV) in Beijing were 0.22 m and 4.36 m$^3$; in Shanghai, they were 0.08 m and 4.65 m$^3$; and in Seattle, they were 0.36 m and 66.55 m$^3$. The average aV was obviously lower than 5 m$^3$ in Beijing and Shanghai. Seattle's aV was significantly higher than these two Chinese cities, about ten times as much, had more carbon sequestration, and played a positive role in urban carbon trading.

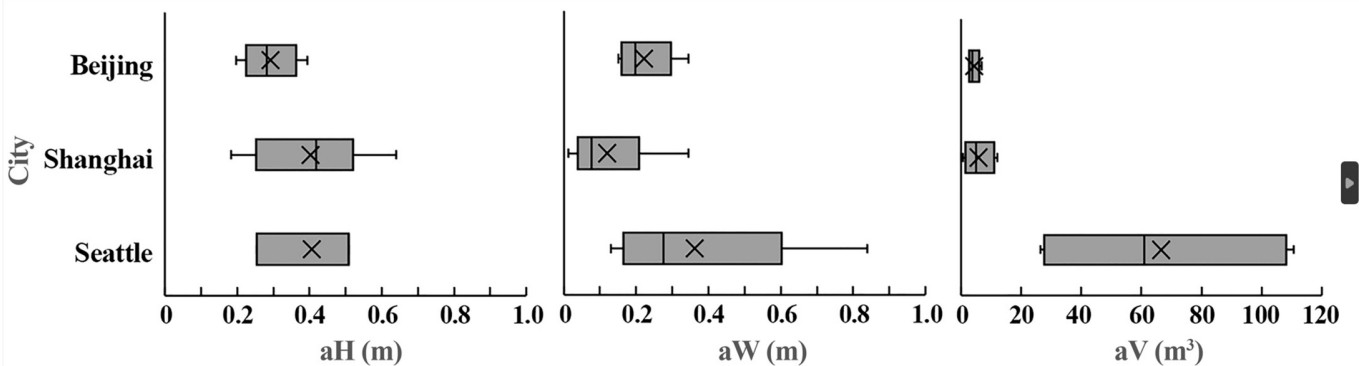

**Figure 4.** Average annual aH, aW, and aV in Beijing, Shanghai, and Seattle.

The crown shapes of sphere and columnar were frequently used in Beijing, Shanghai, and Seattle (Table 4). The average annual tree height increment (aH) of columnar crowns in Seattle was 0.42 m; the aW was 0.44 m; and the aV was 66.89 m$^3$. The aH of columnar crowns in Beijing and Shanghai was 0.26 m; the aW was 0.11 m; and the aV was 4.07 m$^3$, which were significantly different from Seattle. Similarly, the aV of spherical crowns in the two Chinese cities was 6.67 m$^3$, which is about one-tenth of the aV in Seattle. Cone canopies were only present in Beijing and Shanghai, revealing the lowest aV and aW.

**Table 4.** Comparisons of aH, aW and aV in Beijing, Shanghai and Seattle.

| City | Species | Solid Geometry Shape of Tree Crown | aH (m) | aW (m) | aV (m³) |
|---|---|---|---|---|---|
| Beijing, China | *S. japonica* | sphere | 0.25 | 0.34 | 6.86 |
| | *P. tomentosa* | sphere | 0.20 | 0.20 | 2.85 |
| | *F. chinensis* | sphere | 0.39 | 0.15 | 3.84 |
| | *S. babylonica* | sphere | 0.28 | 0.25 | 2.85 |
| | *P. acerifolia* | columnar | 0.33 | 0.17 | 5.42 |
| Shanghai, China | *C. camphora* | sphere | 0.45 | 0.09 | 7.48 |
| | *P. acerifolia* | columnar | 0.18 | 0.05 | 2.71 |
| | *M. glyptostroboides* | cone | 0.39 | 0.01 | 0.68 |
| | *S. saponaria* | sphere | 0.48 | 0.16 | 10.73 |
| | *G. biloba* | cone | 0.64 | 0.07 | 1.68 |
| | *A. buergerianum* | sphere | 0.28 | 0.34 | 12.07 |
| Seattle, USA | *A. nigrum* 'Green Column' | columnar | 0.51 | 0.84 | 110.70 |
| | *F. americana* 'Empire' | columnar | 0.25 | 0.28 | 60.98 |
| | *G. biloba* 'Princeton Sentry' | columnar | 0.51 | 0.20 | 28.98 |
| | *A. saccharum* 'Bonfire' | sphere | 0.51 | 0.37 | 105.63 |
| | *U. parvifolia* 'Emer II' | sphere | 0.25 | 0.13 | 26.44 |

## 4. Discussion

### 4.1. Exploring the Comparable CVs in Different Cities

This study found that the CV in Shanghai (23.93~142.35 m³) was significantly lower than that in Paris, Seattle, and Los Angeles. The relatively small crown diameter of street trees in Shanghai, attributed to high-intensity artificial pruning, poor growing conditions, and a low annual growth rate, may contribute to the lower CV. Taking the most frequent crown shape, the sphere, as an example, the average CV in Shanghai was less than 150 m³ per tree, accounting for about one-fifth of that in the four Western cities. The same situation also occurred for trees with other crown shapes. Compared with trees in Seattle, the aV per street tree in Shanghai also maintained a slower growth trend. The growth of street trees is inevitably affected by tree species replacement, pests, and diseases. The extremely low CV in China has become an issue that urgently needs attention. Decision and policy makers should improve the problem of the low CV of street trees by setting relevant provisions and supervisory personnel. In addition, more and earlier concerns about the growing conditions of street trees and related factors have been raised in Western countries [54]. Research assessing the current state of greenery and online virtual research is worthy of future study in China [55].

However, tree growth depends on the climate, the distribution of rainfall during the year, the different prunings, the space available along city streets, and pollution; this study only used a rough online method to compare the CV between different cities. Further exploration of the reasons for different CVs and CVIs requires field research and experiments as a supplement, which need to be used in subsequent research.

### 4.2. CV Differences in Multi-City Applications of P. acerifolia

*P. acerifolia* (Platanaceae) is one of the world's most widely used street trees and has earned the reputation of the "King of Road Trees" [56,57]. Its longevity, large shade, and high resistance make it a highly cost-effective tree species, widely planted worldwide [58,59]. France was the first city to use *P. acerifolia* as street trees, which are popularly planted in many Chinese cities such as Shanghai, Wuhan, and Nanjing. In this study, *P. acerifolia* was selected for street trees in London, Paris, Shanghai, and Beijing, so comparing the data from the four cities can reflect the differences in CV.

There were significant differences in the CVs of *P. acerifolia* in Beijing, Shanghai, London, and Paris (Table 3). The CVs in London and Paris were 591.99 m³ and 642.50 m³, which were more than five times that in Shanghai. Sparse foliage and slow growth may be

the causes of this [60]. This is closely related to inappropriate pruning and poor habitat conditions in urban greening management and maintenance. In Paris and London, the crown shapes of *P. acerifolia* are mainly pruned in "natural" and "geometric" states [61]. The "natural" state often uses elevated pruning to ensure regular vehicular and pedestrian traffic without eliminating excess branches, while the latter is manually pruned to create geometric shapes to increase its landscape effect [62]. Either way, the crown can remain in a more intact state. Because of the extended planting period, the CV is large, so the capacity for carbon sequestration is strong. In Shanghai, however, the pruning pattern becomes one of the key limiting factors for CV, such as pursuing neatness blindly, chaotic crown structures, serious crown deviations, excessively long upright branches, and large branches with rotten cuttings, leading to an unhealthy growth of pendant trees into open-center crowns [63]. The life expectancy of urban trees can be shortened because of improper pruning techniques that can cause trees to be susceptible to disease and insect damage [17].

### 4.3. Feasibility and Limitations of Virtual Research

The concept of living vegetation volume (LVV) is an indicator for quantifying vegetation amount [64]. The crown volume fraction (CVF) is defined as the volume occupied by tree crowns within a street canyon section and is used to study traffic pollutant concentrations at the pedestrian level [65]. LVV, CVF, and CV are all indices measuring the volume of vegetation, and it is clear that three-dimensional greening indices have advantages when evaluating the amount of green space. In this context, the CV can better embody this concept. The CV can be calculated in several ways. Traditional field measurement methods are often labor-intensive, time-consuming, and limited by spatial accessibility [66]. LiDAR is an active remote sensing technology that can be used to record three-dimensional information about objects [67]. The potential of ALS and TLS and the application of laser scanning using a UAV on landscape and regional scales are well recorded in the literature [46,68,69].

Although this LiDAR measurement method can provide high-precision data, it requires high technological support and funding costs. A new low-cost method for virtual studies, the PCAD, was used in this study, and it also allows for the dynamic monitoring of morphological characteristics and the replacement of street trees. However, some deficiencies were also found during the research process, mainly due to the errors caused by human operation. Reducing errors due to manual operation and improving efficiency are issues that need to be solved. In the future, existing open resources will be used and integrated to build a more extensive-scale database, and the PCAD will be applied to more cities around the world to assess the CVI, which is related to the contribution to carbon sequestration and tree health of urban trees.

### 5. Conclusions

This study compared the CV per street tree and its yearly increment between two Chinese cities and four Western cities by using the new method of the PCAD and online datasets. The street trees in China showed lower CVs, ranging from one-half to one-tenth of those observed in the Western cities, which is closely related to inappropriate pruning and poor habitat conditions in urban greening management and maintenance. The application of the PCAD provides new ideas for urban tree management and carbon sequestration evaluation and a basis for government decision making in areas with low CVIs. Compared with other methods that require high levels of technological support and funding costs, the PCAD has the advantages of free resource collection, low costs, and simple operation. However, local climate conditions should not be neglected as environmental factors. Further exploration should be conducted for the potential causes of different CVs and CVIs, which requires field research and experiments as a supplement. The automatic detection of street trees using deep learning-based objects remains challenging [70]. Our current research represents just the initial phase, and we are committed to further refining and expanding upon our method, resulting in an improved iteration of the PCAD. Future research will focus on enhancing its speed, efficiency, and cost effectiveness in this regard, increasing the

identification of tree age and health, and analyzing environmental indicators. We intend to apply this method to assess CV and CVI in additional countries and address global tree health and carbon sequestration concerns on a global scale.

**Author Contributions:** Conceptualization, H.W.; methodology, H.W. and C.G.; software, D.W. and L.X.; validation, D.W. and L.X.; formal analysis, D.W. and L.X.; investigation, C.G.; resources, Y.H.; data curation, Y.H. and J.Q.; writing—original draft preparation, C.G. and H.W.; writing—review and editing, Y.H. and J.Q.; visualization, C.G.; supervision, Y.H.; project administration, Y.H.; funding acquisition, J.Q. All authors have read and agreed to the published version of the manuscript.

**Funding:** This research was funded by the Shanghai Municipality Science and Technology Commission, grant number 21DZ1202003; the Shanghai Landscaping and City Appearance Administrative Bureau, grant number G212409; and the Shanghai Engineering Research Center of Plant Germplasm Resources, grant number 17DZ2252700.

**Data Availability Statement:** The data presented in this study are available on request from the corresponding author. The data are not publicly available due to privacy.

**Acknowledgments:** We are grateful to Ao Wu for the language editing. We also thank the editors and three anonymous reviewers for their valuable suggestions.

**Conflicts of Interest:** The authors declare no conflicts of interest.

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
