# Peer review of "Comparison of Crown Volume Increment in Street Trees among Six Cities in Western Countries and China"

_horticulturae, doi:10.3390/horticulturae10030300_

Round 1

Reviewer 1 Report

Comments and Suggestions for Authors

The study entitled:"Comparison of crown volume of street trees among the six cities in Western countries and China", by Guo et al. focuses on tree crown volume (CV) as a vital indicator for assessing the ecological quality of urban environments and the carbon sequestration potential of trees. A new, cost-effective method called PCAD was utilized to obtain CV data in China, while primary data on street trees from four Western cities were sourced from online datasets. The paper is generally well-written, well-organized, and contains several important information. The paper is novel and can be accepted with some minor revisions.

Strengths of the current study: 

* The inclusion of cities from both China and Western countries allows for a diverse comparison, considering varying urban planning approaches and green space development levels.

* The methodology for calculating CV and its increments is based on established geometric formulas, incorporating crown diameter, height, and tree species-specific characteristics. This can be considered as a strength point of the study.

Weaknesses of the study:

* Selecting Beijing and Shanghai solely based on their urban green space planning may introduce bias, as other factors influencing tree growth, such as climate and soil conditions, are not explicitly considered.

* Relying on Street View for cities lacking open datasets may introduce limitations, such as restricted access to certain areas and potential inconsistencies in tree identification and measurement, which could affect data reliability. Can the authors give a detailled explanation on this matter.

In the abstract section: remove the numbers (1), (2) ... and so on. Although the abstract gives a general overview of the study, it is devoid of precise information about the technique and statistical analysis that were used.

Line 20 : remove "showed" or "exhibited" to enhance clarity.

Author Response

March 16, 2024

Dear,

Thank you for the chance to revise and improve our manuscript, and we are very grateful to the Editors and anonymous reviewers for their comments and advice on the manuscript titled "Comparison of crown volume of street trees among the six cities in Western countries and China" (horticulturae-2865809).

All the responses to the questions of the Editor and reviewers can be found with underlines in the following. We submitted two copies of the manuscript. One is the tracked copy with revision marks, and the other is the clear copy. The number of lines mentioned in our responses refers to the lines in the tracked copy.

The focus of the revision is to address some issues of unclear content expression and increase the discussion of the relationship between CV, tree health, and carbon storage. Discussion on the impact of the environment, tree age, and other factors on the results was also added to the article. In addition, we revised the title based on the reviewers' comments.

We are looking forward to hearing from you as soon as possible.

Sincerely,

Hongbing Wang, PhD

College of Life Science, Shanghai Normal University

No. 100 Haisi Road, Fengxian District, Shanghai, China, P.C. 201418

Tel: +86 21 5122689

Email: whb0236@shnu.edu.cn

Review of horticulturae-2865809

Comparison of crown volume of street trees among the six cities in Western countries and China

Journal: Horticulturae

The study entitled: "Comparison of crown volume of street trees among the six cities in Western countries and China", by Guo et al. focuses on tree crown volume (CV) as a vital indicator for assessing the ecological quality of urban environments and the carbon sequestration potential of trees. A new, cost-effective method called PCAD was utilized to obtain CV data in China, while primary data on street trees from four Western cities were sourced from online datasets. The paper is generally well-written, well-organized, and contains several important information. The paper is novel and can be accepted with some minor revisions.

Strengths of the current study: 

* The inclusion of cities from both China and Western countries allows for a diverse comparison, considering varying urban planning approaches and green space development levels.

* The methodology for calculating CV and its increments is based on established geometric formulas, incorporating crown diameter, height, and tree species-specific characteristics. This can be considered as a strength point of the study.

Weaknesses of the study:

* Selecting Beijing and Shanghai solely based on their urban green space planning may introduce bias, as other factors influencing tree growth, such as climate and soil conditions, are not explicitly considered.

Response: Thanks for your comments. As per your suggestion, different environmental factors such as climate can have an impact on CV, so we have also included this viewpoint in our discussion (lines 321-326). The PCAD and online databases we used were only able to obtain limited data. Our study compared the CV of street trees in Chinese cities and Western cities from an overall perspective. The research needed to be improved, further exploration of the reasons for different CV and CV increment requires field research and experiment as a supplement, which needs to be improved in subsequent research (lines 389-390, 395-396).

* Relying on Street View for cities lacking open datasets may introduce limitations, such as restricted access to certain areas and potential inconsistencies in tree identification and measurement, which could affect data reliability. Can the authors give a detailled explanation on this matter.

Response: Thank you for your suggestion. The London and Los Angeles data we downloaded from online datasets lacked stem height, which was used to calculate crown height. The data for Paris and Seattle lacked crown diameter and stem height. Referring to Guo et al., 2023 (https://doi.org/10.1016/j.ufug.2023.128029), we used Street View to measure the missing data based on the tree location information given in the online datasets. Based on a reference of known dimensions, the measurement of stem height and crown diameter was conducted for each tree species. The number of sampling trees was below 100 when the total number of trees was lower than 100, and all of the trees would be measured; the sampling trees should be randomly selected for measurement, and the sampling number was 100 when the total number of trees was more than 100. If there are no explicit images in some areas or restricted access, we can measure the same tree species in other locations as long as the number of sample trees meets the above requirements. We also acknowledge errors in tree identification and measurement using this method. However, some studies have confirmed the feasibility of this method, and we have added citations in the article (lines 149, 175). Our current research represents just the initial phase, and we are committed to further refining and expanding upon our method, an improved iteration of PCAD. Future research will focus on enhancing speed, efficiency, and cost–effectiveness, increasing identification of tree age and health, and analyzing the environmental indicators.

In the abstract section: remove the numbers (1), (2) ... and so on. Although the abstract gives a general overview of the study, it is devoid of precise information about the technique and statistical analysis that were used.

Response: Thank you for your opinion. We have added the description of the technique and statistical analysis that were used (lines 18-19).

Line 20 : remove "showed" or "exhibited" to enhance clarity.

Response: Thank you for your opinion. We have modified (line 22).

Reviewer 2 Report

Comments and Suggestions for Authors

Paper ID: horticulturae-2865809-peer-review

Paper Title: Comparison of crown volume of street trees among the six cities in Western countries and China

The work is well structured. The authors followed standard procedures to collect the data and derive the results. However, for further improvement, I would like to bring the following issues to the attention of the authors:

Section-specific comments:

ABSTRACT SECTION

In the abstract, the method section is not well prepared. There are no known data sources for data analysis using PCAD. Also, the abbreviation, PCAD should be used in full terms in its first appearance.

Clearly state the objective of your study.

INTRODUCTION SECTION

The introduction section is well organized. Only the authors need to state the uniqueness of their studies compared to those previously available. Also, the time of study is not defined as urban trees’ numbers and types are dynamic across time.

METHODS SECTION

Line numbers from 142-146, the information can be put into a table.

Is the approach accurate for species identification?

What is your rationale for downloading images for 2019, 2010, 2020, and 2021? The interval is not uniform, and the sample is not equal. There are more images for 2010 than for the rest of the study years.

Lines 168–169, for Beijing and Shangai, the image you downloaded was for 2010, 2019, 2020, and 2021, but in the equation, you used 2011 and 2022, in which both years were not covered in your downloaded images.

The interval between the sample plots is defined (3 km), but how large is the area of the sample plot?  

The time for data collection in Paris is not defined.

RESULTS

The results are well stated, but where are the CV differences for each species identified?

The present way of presenting the results is less attractive. Can you use graphs for some of them?

DISCUSSION SECTION

This part is adequately discussed, but what is missing:

1. The implications of your study for decision- and policy-makers

CONCLUSION

For your conclusion on the lower CV in China, you suggested ‘inappropriate pruning and poor habitat conditions in urban greening management and maintenance’ as a possible cause. But, in your study, no method was mentioned to cover pruning and other urban tree management methods. What about the nature of the species? The local climate effects? The spacing between the trees? Many other factors may contributed to the lower CV in China.

Good luck!

Author Response

March 16, 2024

Dear,

Thank you for the chance to revise and improve our manuscript, and we are very grateful to the Editors and anonymous reviewers for their comments and advice on the manuscript titled "Comparison of crown volume of street trees among the six cities in Western countries and China" (horticulturae-2865809).

All the responses to the questions of the Editor and reviewers can be found with underlines in the following. We submitted two copies of the manuscript. One is the tracked copy with revision marks, and the other is the clear copy. The number of lines mentioned in our responses refers to the lines in the tracked copy.

The focus of the revision is to address some issues of unclear content expression and increase the discussion of the relationship between CV, tree health, and carbon storage. Discussion on the impact of the environment, tree age, and other factors on the results was also added to the article. In addition, we revised the title based on the reviewers' comments.

We are looking forward to hearing from you as soon as possible.

Sincerely,

Hongbing Wang, PhD

College of Life Science, Shanghai Normal University

No. 100 Haisi Road, Fengxian District, Shanghai, China, P.C. 201418

Tel: +86 21 5122689

Email: whb0236@shnu.edu.cn

Review of horticulturae-2865809

Comparison of crown volume of street trees among the six cities in Western countries and China

Journal: Horticulturae

Paper ID: horticulturae-2865809-peer-review

Paper Title: Comparison of crown volume of street trees among the six cities in Western countries and China

The work is well structured. The authors followed standard procedures to collect the data and derive the results. However, for further improvement, I would like to bring the following issues to the attention of the authors:

Section-specific comments:

ABSTRACT SECTION

In the abstract, the method section is not well prepared. There are no known data sources for data analysis using PCAD. Also, the abbreviation, PCAD should be used in full terms in its first appearance.

Response: Thank you for your opinion. We have modified (lines 16-17).

Clearly state the objective of your study.

Response: Thank you for your opinion. We have modified (lines 24-28). The purpose of the research was to verify the operability of PCAD and compare the CVI of different cities.

INTRODUCTION SECTION

The introduction section is well organized. Only the authors need to state the uniqueness of their studies compared to those previously available. Also, the time of study is not defined as urban trees’ numbers and types are dynamic across time.

Response: Thank you for your opinion. We added the description of innovation points in lines 111-113. The uniquenesses of this study are as follows: first, compared with methods such as UAV and LiDAR equipment to calculate CV, this study used a new method named PCAD, whose credibility had been verified in previous papers (https://doi.org/10.1016/j.ufug.2023.128029); secondly, few studies have paid attention to CVI, and the research methods are relatively single. As you said, we can see dynamic changes in the number of street trees by comparing satellite images at different times. We mentioned this in the 2.3 section (line 178). The aH, aW, and aV of the remaining street trees were analyzed by excluding the changed trees. At the same time, we have added Table 1 to list the date of data sources in the six cities so that readers can read more clearly.

METHODS SECTION

Line numbers from 142-146, the information can be put into a table.

Response: Thank you for your opinion. We put the information of the satellite images into the table (line 178).

Is the approach accurate for species identification?

Response: Identification of tree species through street view is highly accurate, and we have added references here (line 149,175).

What is your rationale for downloading images for 2019, 2010, 2020, and 2021? The interval is not uniform, and the sample is not equal. There are more images for 2010 than for the rest of the study years.

Response: Satellite images in Google Earth do not include all dates. In the study, we tried to compare the data of 2010 with that of 2020. Unfortunately, at some sample points, we could not download the images of these two years, or the images of these two years needed to be clarified.  Therefore, we selected images from similar years (2009, 2019, and 2021) for analysis.

Lines 168–169, for Beijing and Shangai, the image you downloaded was for 2010, 2019, 2020, and 2021, but in the equation, you used 2011 and 2022, in which both years were not covered in your downloaded images.

Response: Thank you for your opinion. We have modified (lines 193-198). In our previous reply, we explained the reasons for using 2009 data instead of 2010, 2019 and 2021 data instead of 2020. Therefore, we modified the formula. The 2022 in the second formula was the time of the data in Seattle, which was from open and free online datasets. This is explained in Table 1.

The interval between the sample plots is defined (3 km), but how large is the area of the sample plot?  

Response: Thank you for your opinion. We have modified (line 162). Each plot is a circle with a diameter of 1 km and an area of approximately 3.14 km2.

The time for data collection in Paris is not defined.

Response: The data for Paris in 2021 was sourced from the Paris Open Data website. This is explained in Table 1.

RESULTS

The results are well stated, but where are the CV differences for each species identified?

Response: Thank you for your opinion. We have added the content about CV differences for each species (lines 223-226). Among all 27 tree species, the top three species with the highest CV were Acer saccharum, Cinnamomum camphora, and Fraxinus americana, the top three tree species with the lowest CV were Populus tomentosa, Salix babylonica, and Metasequoia glyptostroboides.

The present way of presenting the results is less attractive. Can you use graphs for some of them?

Response: Thank you for your opinion. We have added Figure 3 and Figure 4.

DISCUSSION SECTION

This part is adequately discussed, but what is missing:

  1. The implications of your study for decision- and policy-makers

Response: Thank you for your opinion. We have added this content to the manuscript (lines 315-317).

CONCLUSION

For your conclusion on the lower CV in China, you suggested ‘inappropriate pruning and poor habitat conditions in urban greening management and maintenance’ as a possible cause. But, in your study, no method was mentioned to cover pruning and other urban tree management methods. What about the nature of the species? The local climate effects? The spacing between the trees? Many other factors may contributed to the lower CV in China.

Response: Thank you for your opinion. As per your suggestion, different environmental factors such as climate can impact on CV, so we have included this viewpoint in our Discussion section (lines 321-326, 395-396). Our research needs to improve. Further exploration of the reasons for different CV and CVI requires field research and experiment as a supplement, which needs to be improved in subsequent research (lines 389-390).

Reviewer 3 Report

Comments and Suggestions for Authors

Although the main purpose of the manuscript is to suggest a new, low-cost method for determining tree canopy volume, it will be important for the authors to define or clarify several points. The topic is important for the area of urban forestry and arboriculture, however, it will be important to review other references where they point out that the volume per se is not sufficient to determine health or even carbon storage; At least, it should be noted in the introduction. In addition to the fact that this section provides a lot of data, it seems to me that the authors should focus on clearly pointing out the connection with the crown volume. Subsequently, they point out that the increase in volume will be analyzed, a variable that can better provide more information about the objective of the work, but curiously it is not mentioned in the title of the manuscript. In the data analysis, it must be clear which variables were analyzed using parametric statistics and which were not. Within the results, they are a bit confusing, since, although cities share some species, it is well known that due to environmental conditions, the values will be different and if not, where is the effect of tree age. The above generates noise in the analysis and conclusions of the data. In the discussion, if it is based on a result, it will be important that the methodology indicates how the data was obtained or, where appropriate, if it is the result of a comparison with previous research. Once the points have been clarified, it will be important for the authors to review the conclusions. PDF file is attached for additional comments.

Comments on the Quality of English Language

The English quality language seems ok

Author Response

March 16, 2024

Dear,

Thank you for the chance to revise and improve our manuscript, and we are very grateful to the Editors and anonymous reviewers for their comments and advice on the manuscript titled "Comparison of crown volume of street trees among the six cities in Western countries and China" (horticulturae-2865809).

All the responses to the questions of the Editor and reviewers can be found with underlines in the following. We submitted two copies of the manuscript. One is the tracked copy with revision marks, and the other is the clear copy. The number of lines mentioned in our responses refers to the lines in the tracked copy.

The focus of the revision is to address some issues of unclear content expression and increase the discussion of the relationship between CV, tree health, and carbon storage. Discussion on the impact of the environment, tree age, and other factors on the results was also added to the article. In addition, we revised the title based on the reviewers' comments.

We are looking forward to hearing from you as soon as possible.

Sincerely,

Hongbing Wang, PhD

College of Life Science, Shanghai Normal University

No. 100 Haisi Road, Fengxian District, Shanghai, China, P.C. 201418

Tel: +86 21 5122689

Email: whb0236@shnu.edu.cn

Review of horticulturae-2865809

Comparison of crown volume of street trees among the six cities in Western countries and China

Journal: Horticulturae

Although the main purpose of the manuscript is to suggest a new, low-cost method for determining tree canopy volume, it will be important for the authors to define or clarify several points. The topic is important for the area of urban forestry and arboriculture, however, it will be important to review other references where they point out that the volume per se is not sufficient to determine health or even carbon storage; At least, it should be noted in the introduction.

Response: Thank you for your opinion. In the Introduction section, we added references about the relationship between CV and tree health, even carbon storage (lines 51-68). A healthy tree with a larger crown diameter usually has a larger quantity of leaves. Air pollution capture, rainfall retention, and shade provision are directly related to the surface area of the leaves and branches within the tree crown and the architecture of the crown. CV becomes the key to the estimation of biomass and carbon stored.

In addition to the fact that this section provides a lot of data, it seems to me that the authors should focus on clearly pointing out the connection with the crown volumezai. Subsequently, they point out that the increase in volume will be analyzed, a variable that can better provide more information about the objective of the work, but curiously it is not mentioned in the title of the manuscript.

Response: Thank you for your opinion. We have modified (lines 43-68).

In the data analysis, it must be clear which variables were analyzed using parametric statistics and which were not.

Response: We have modified (lines 210-213). The annual tree height increment (aH) was analyzed using parametric statistics and the average tree height (H), average crown diameter (W), average crown volume (CV), annual crown diameter (aW), and annual crown volume (aV) were not.

Within the results, they are a bit confusing, since, although cities share some species, it is well known that due to environmental conditions, the values will be different and if not, where is the effect of tree age. The above generates noise in the analysis and conclusions of the data. In the discussion, if it is based on a result, it will be important that the methodology indicates how the data was obtained or, where appropriate, if it is the result of a comparison with previous research. Once the points have been clarified, it will be important for the authors to review the conclusions.

Response: We appreciate your perspective. We acknowledge the omission of tree age classification in our study and have included this information in the Conclusion section for transparency (lines 395-396). Our study compared the CV of street trees in Chinese cities and Western cities from an overall perspective and can provide some help for research in urban ecology. We will focus on solving this problem in the next experiment, increasing the identification of tree age and analysis of environmental indicators.

In 4.1-4.2 of the Discussion section, we discussed the application of PCAD in Beijing and Shanghai and compared the CV and CVI of these two cities with Western cities. In 4.3, we compared PCAD with other methods for calculating CV. The Conclusion section is also divided into two aspects: the application results of PCAD in Beijing and Shanghai (lines 377-382) and the comparison of PCAD with other methods (Lines 384-386). We also illustrated the shortcomings and prospects of PCAD (Lines 386-396).

 PDF file is attached for additional comments:

Lines 36-44 These are important points of urban trees but it will be important to focus them on the volume of the canopy so as not to dissipate ideas.

Response: Thank you for your opinion. We have deleted some content about the importance of urban trees. At the same time, we have added references about the relationship between CV and tree health, even carbon stock, in the Introduction section (lines 43-68).

Lines 48-49 please review: Morales-Gallegos, L.M.;Martínez-Trinidad, T.; Hernández-dela Rosa, P.; Gómez-Guerrero, A.;Alvarado-Rosales, D.;Saavedra-Romero, L.d.L. Tree HealthCondition in Urban Green AreasAssessed through Crown Indicatorsand Vegetation Indices. Forests 2023,14, 1673. https://doi.org/10.3390/f14081673

Response: Thank you for your reference. We have added reference here (lines 54-55) and quoted the views about “Crown assessment can be used as an indicator of health conditions in forest trees”.

Line 82 vigor

Response: We have modified.

Line 94 then, it should be added in the manuscript title and and emphasized it in the summary

Response: Thank you for your opinion. We have modified (lines 390, 397).

Line 95 As this is the first time it is mentioned in the manuscript, the full name must be written

Response: We have modified (line 115).

Line 104 sequestration and tree health

Response: We have modified (lines 124-125).

Line 182 explaing what data?

Response: We have modified (line 210). This explained the annual tree height increment (aH).

Line 185 which variables?

Response: We have modified (line 213). This explained the average tree height (H), average crown diameter (W), average crown volume (CV), annual crown diameter (aW), and annual crown volume (aV).

Line 192 How to determine the weight of the site? The differences were to be expected due to the different environmental growing conditions - without taking into account the space conditions in each city.

Response: We appreciate your perspective. Our study compared the CV of street trees in Chinese cities and Western cities from an overall perspective and can provide some help for research in urban ecology. In the Discussion section, we also mentioned that “the tree growth depends on the climate, the distribution of rainfall during the year, the different pruning, the space available along city streets and pollution, this study only used a rough online method to compare the CV between different cities” (lines 321-324). We will focus on solving this problem in the following experiment (lines 388-390, 394-396).

Line 197 But how to know that both types of trees are the same age or at least similar?

Response: We acknowledge the omission of tree age classification in our study and have included this information in the Conclusion section for transparency (lines 394-396). It is worth noting that trees with smaller crowns tend to exhibit smaller CVI, but we have taken the overall context into account. In cases where a city predominantly plants young trees, its CV for street trees may appear smaller compared to other cities. We will focus on solving this problem in the next experiment.

Line 267 Since only the volume is measured in this case and not the increase, what is the effect of age?

Response: Thank you for your opinion. We acknowledge the omission of tree age classification in our study. The comparison here illustrates the differences in CV across multiple cities simultaneously (line 304). Regarding the question about the tree age, please see the previous answer.

Line 272 How was it measured?

Response: Thank you for your opinion. We have removed this inappropriate expression.

Line 305 please review: Morales-Gallegos, L.M.;Martínez-Trinidad, T.; Hernández-dela Rosa, P.; Gómez-Guerrero, A.;Alvarado-Rosales, D.;Saavedra-Romero, L.d.L. Tree HealthCondition in Urban Green AreasAssessed through Crown Indicatorsand Vegetation Indices. Forests 2023,14, 1673. https://doi.org/10.3390/f14081673

Response: We have modified (line 344).

Line 341 It is not clear how this is analyzed ?

Response: Thank you for your opinion. We have modified this inappropriate expression. We have added references about the relationship between CV and tree health, even carbon storage, in the Introduction section (lines 51-68). So we changed this sentence to “The CV is positively related to the health and carbon stored potential of trees”.

Line 348 How come they conclude about this but indicate that they are barely going to do it?

Response: We have modified (lines 396-398). We intend to apply this method to assess CV and CVI in additional countries and address global tree health and carbon sequestration concerns globally.

Round 2

Reviewer 2 Report

Comments and Suggestions for Authors

I want to express my gratitude to the authors for considering my feedback and incorporating my suggestions into the revised manuscript. The article has been enhanced and is now suitable for publication.

Reviewer 3 Report

Comments and Suggestions for Authors

I am pleased to note that the authors addressed the comments made in the arbitration process; Therefore, I would not have additional comments on the document.